# [Re] Explaining in Style: Training a GAN to explain a classifier in StyleSpace

## Reproducibility Summary

**Scope of Reproducibility**

StylEx is an approach for classifier-conditioned training of a StyleGAN2 [6], intending to capture classifier-specific attributes in its disentangled StyleSpace [15]. Attributes can be adjusted to generate counterfactual explanations of the classifier decisions. StylEx is domain and classifier-agnostic, while its explanations are claimed to be human-interpretable, distinct, coherent and sufficient to produce flipped classifier decisions. We verify these claims by reproducing a selection of the experiments in the paper.

**Methodology**

We verified a selection of the experimental results on the code available by the authors. However, a significant part of the training procedure, network architecture and hyperparameter configurations were missing. As such, we reimplemented the model and available TensorFlow code to PyTorch, to enable easier reproducibility on the proposed case studies. All experiments were run in approximately 20-50 GPU hours per dataset, depending on the batch size, gradient accumulation and GPU.

**Results**

We verified that the publicly available pretrained model has a 'sufficiency' measure within 1% of the value reported in the paper. Additionally, we evaluate the *Fréchet inception distance* (FID) scores of images generated by the released model. We show that the FID score increases with the number of attributes used to generate a counterfactual explanation. Custom models were trained on three datasets, with a reduced image dimensionality ($64^2$). Additionally, a user study was conducted to evaluate the distinctiveness and coherence of the images. We report a significantly lower accuracy on the identification of the extracted attributes and 'sufficiency' scores on our model.

**What was easy**

It was easy to run the provided Jupyter Notebook, and verify the results of the pretrained models on the FFHQ dataset. Extending an existing StyleGAN2 implementation to fit this study was relatively easy.

**What was difficult**

Reproducing the experiments on the same scale as the authors, as well as the development of the full training procedure, model architecture and hyperparameters, particularly due to underspecification in the original paper. Additionally, the conversion of code from Tensorflow to PyTorch.

**Communication with original authors**

We corresponded with the first author of the paper through several emails. Through our mail contact, additional details were released on the network architecture, the training procedure and the hyperparameter configurations.

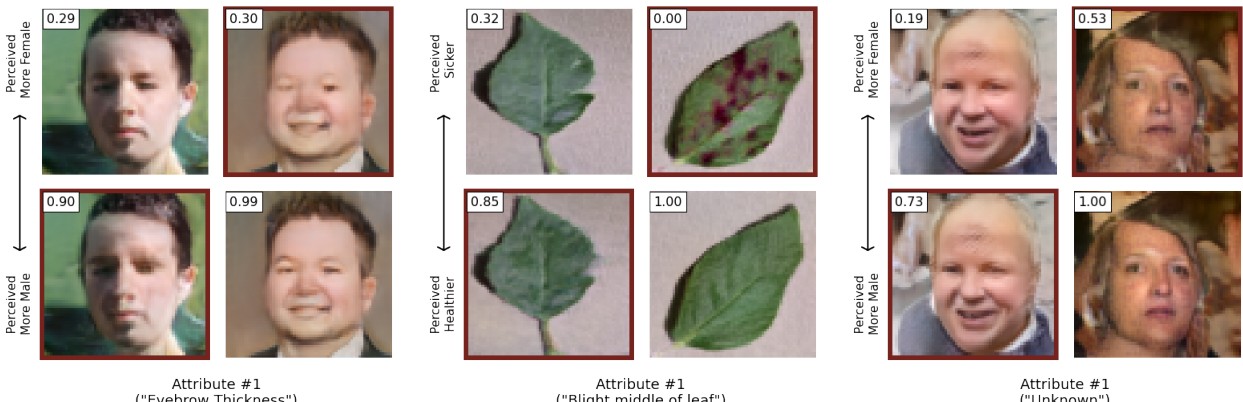

Figure 1: Top-1 automatically detected attributes for perceived-gender classifiers (left: version 1, right: version 2) and perceived-health of leaves classifiers (middle). Similarly to the original paper, the counterfactual images are marked by a frame. Displayed probabilities correspond to the person being male for perceived gender and the leaf being healthy for perceived health of leaf. More attributes can be found in the appendix.

# 1 Introduction

Existing post-hoc visual explainability measures, such as heatmaps[13], can highlight regions that influence the decision. However, they do not visualize non-spatial localized attributes nor do they indicate how these areas may be changed to influence the classification. Counterfactual explanations, which are statements of the form "Had the input $\mathbf{x}$ been $\tilde{\mathbf{x}}$, the classifier output would have been $\tilde{\mathbf{y}}$ instead of $\mathbf{y}$", have been proposed as an alternative which both specifies the important features and naturally explains how it can be altered to achieve an alternative outcome.

As such, these explanations are promising as they can provide a suggestive recourse to non-domain experts in a machine learning-based decision system. The effectiveness of these methods strongly depend on the intuitive difference that humans observe; therefore one of the primary objectives is to find these attributes. Secondary objectives involve the visualization and control of the impact of these features on the classifier output.

In this work, we reproduce the paper 'Explaining in Style: Explaining a GAN in StyleSpace' [8]. The paper proposes a novel method for explaining the classification of a given image, by altering discovered human-interpretable features discovered to affect the classification output. We reimplemented the model in PyTorch together with the training procedure, as the original TensorFlow implementation lacked the training procedure code. We performed training on the FFHQ and PlantVillage dataset using a lower resolution. Using our own implementation, we check whether the results are consistent with the descriptions provided in the paper. We strengthen this with the addition of a human-grounded evaluation of the generated images. Additionally, we used the FID measure to evaluate the image quality of the counterfactual generated images.

# 2 Scope of Reproducibility

The StylEx model, in addition to the *AttFind* algorithm defined in the paper, is presented as a viable option for generating counterfactual explanations of black-box classifiers. The StylEx model aims to make individual classifier-relevant, through a novel training procedure which is outlined in 3.

As no benchmark metrics exist to evaluate and assess attribute-based counterfactual explanations, the authors propose three evaluation criteria themselves: 1) visual coherence, 2) distinctness and 3) 'effect of attributes on classification' (sufficiency). We reformulate these criteria as the main claims of the paper in the following manner:

1. **Visual Coherence:** Attributes detected by StylEx should be clearly identifiable by humans.

2. **Distinctness:** The attributes extracted by StylEx should be distinct.

3. **Sufficiency:** Changing attributes should result in a change of classifier output, where changing multiple attributes has a cumulative effect.

## 3 Methodology

To evaluate claim 1 and 2, the authors conduct a user study in two parts. To evaluate claim 3, they study the percentage of flipped classifications when modifying top-$k$ (in their case $k = 10$) attributes. To reproduce these claims, we conduct the same experiments, albeit at a lower dimensionality of $64^2$. The complex network architecture of StyleGAN, as well as the encoder both require a significant number of training epochs until its convergence and thus, training these at the full resolution of $256^2$ is extremely computationally expensive.

We verify sufficiency scores of the released model, by making use of the supplied Jupyter Notebook. However, several crucial elements were missing, which included details on the training procedure the omission of hyperparameter configurations and details on the optimization procedure. As such, we ported the available TensorFlow code to PyTorch, to enable easier reproducibility on the proposed case studies.

We reimplemented the StylEx model in PyTorch, using an open-source StyleGAN2 implementation as a starting point[1].

For running our code, we have made use of an NVIDIA GTX 1080 Ti, RTX 2070 Super and a laptop RTX 3060 graphics card, running on different machines. In the conduction of the user study, we have made use of the online survey tool Qualtrics [1].

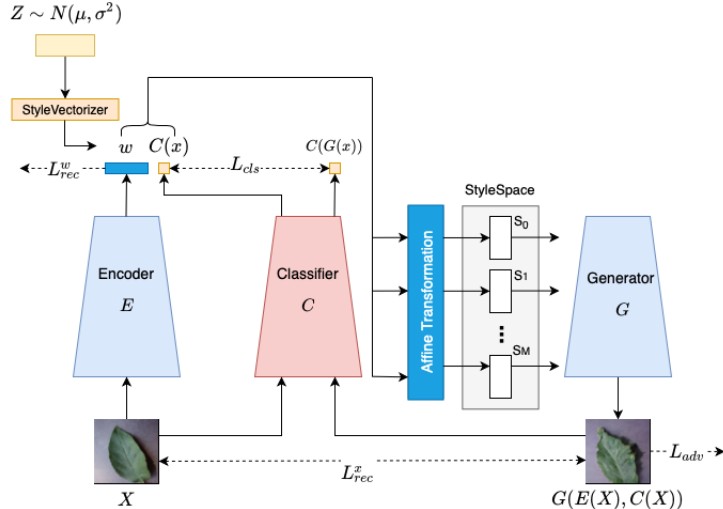

Figure 2: **StylEx network architecture:** with the respective classifier $C$, generator $G$, discriminator $D$ and encoder $E$. For clarification we have slightly adapted the visualization to include the StyleVectorizer which obtains the latent vector $w$ from $z$ [5], after learning that the authors have used alternating training 1

### 3.1 Model descriptions

In addition to a pretrained classifier $C$, StylEx is comprised of three trainable elements, which are a 1) generator $G$, 2) a discriminator $D$ and 3) an encoder $E$. The $D$ and $G$ follow the StyleGAN2 architecture, with minor alterations to $D$ which will be explained below. Figure 2 provides an overview of the network architecture.

Some design details were unspecified or omitted in the original paper. We contacted the authors to provide clarification on these issues, which are stated as follows:

1. StylEx is trained using both encoder input and noise input transformed through StyleGAN2's mapping network, using alternating steps;

2. The output of $D$ is a weighted sum of the 2-dimensional output of its last layer with the input probabilities of the 1) original image if using the encoder, 2) randomly sampled image if using noise input;

3. $\mathcal{L}_{rec}$ and $\mathcal{L}_{cls}$ are only calculated during the generator training steps.

---

[1]https://github.com/lucidrains/stylegan2-pytorch

85   The GAN is trained jointly with the encoder, which embeds an image into the $W$ latent space of StyleGAN2, forming a
86   latent vector $w$. A recent observation by [14] highlighted the disentanglement of this space (appropriately called, the
87   StyleSpace) that is used to extract classifier-specific attributes. Logits of the original image $C(x)$ are then appended
88   to $w$, to condition the training on classifier inputs. The current architecture includes a StyleVectorizer that obtains
89   the latent vector $W$ from $Z$, which is sampled from a normal distribution. In alternating steps the generator was fed
90   input from the encoder and input from the StyleVectorizer mapping network [5]. The original authors noticed a slight
91   improvement in image quality using alternating training, compared to only using the encoder input.

92   We note that we used two slightly different implementation choices for training our models. The first implementation
93   does not include the discriminator change mentioned above, while the second implementation does and uses probabilities
94   instead of logits for concatenation to $w$. We call these two choices 'Model 1' and 'Model 2' in results on datasets where
95   we have trained both. We additionally noted that the MobileNet classifier 'Model 1' was trained with did not perform
96   well on the faces. This is why, for both faces models, a ResNet classifier was used to perform the AttFind algorithm.

97   This expanded latent vector $w$, either obtained by the encoder or StyleVectorizer, is passed on to the StyleGAN2, where
98   it is transformed into the StyleSpace by a set of concurrent affine transformations to style vectors $s_0, ..., s_n$. These style
99   vectors are used to generate novel images, that aim to reconstruct the original image as closely as possible. Several
100  losses are used to quantitatively assess the convergence of the training procedure. The cumulative training loss for the
101  algorithm is a sum of losses, denoted as follows:

$$\text{StylEx}_{\text{Loss}} = \mathcal{L}_{adv} + \mathcal{L}_{reg} + \mathcal{L}_{rec} + \mathcal{L}_{cls}. \tag{1}$$

102  A logistic adversarial loss [2] $\mathcal{L}_{adv}$ is used as in standard GAN training, followed by the regularization loss $\mathcal{L}_{reg}$, as
103  described in the original StyleGAN [6] paper. The reconstruction loss $\mathcal{L}_{rec}$ is given by the sum of $\mathcal{L}_{rec}^x + \mathcal{L}_{rec}^w + \mathcal{L}_{LPIPS}$,
104  where the first two terms are the **L1** distance between original and reconstructed input, and the original and reconstructed
105  $w$ latent vector, respectively. The $\mathcal{L}_{LPIPS}$ term is the LPIPS distance between original and reconstructed input, as
106  described in [17]. This loss ensures that reconstructed images resemble the original input as close as possible, to
107  serve as an input for generating counterfactual examples. The classifier loss is defined as the Kullback-Leibler
108  divergence between the original input image $X$ and the generated new image $G(E(X), C(X))$, defined as follows:
109  $\mathcal{L}_{cls} = D_{KL}[||C(x')||C(x)]$. This loss ensures that the generator does not disregard image attributes that are important
110  for the classification.

111  To extract classifier-specific attributes, the *AttFind* algorithm is proposed in the paper. As input, it takes the trained
112  model $D$ and a set of $N$ images whose predicted label do not match the target label $y$. For each class label, *AttFind*
113  encodes the images and iteratively tries to find a set $S_y$ of $M$ style coordinates that represent the largest possible shift to
114  the opposing class. Next to this, it finds the set of directions $D_y \in 1^M$ that indicate to which class the direction needs
115  to be adjusted to flip the classifier decision. In each iteration, it considers all style coordinates $K$ and determines the
116  coordinate with the largest effect. All images where changing this coordinate results in a large effect on their probability
117  are removed from the iteration. The process is repeated until no images are left, or until $M$ attributes are found.

118  ## 3.2   Datasets

119  We reproduce a selection of the findings of the authors on two of the given datasets in our PyTorch re-implementation:

120      1. **CelebA [9]** The original Large-scale CelebFaces Attributes (CelebA) dataset[2] contains 200000 image entries,
121         each containing 40 attribute annotations. We have trained classifiers on both the gender and age attribute.
122      2. **FFHQ [11]** The original Flickr-Faces-HQ dataset containing 70000 images of human faces. This dataset
123         was used for StylEx training, while the pretrained classifier was trained on the CelebA dataset, following the
124         procedure of the original paper.[3]
125      3. **Plant-Village:** This dataset contains 54303 entries, with 38 categories. This dataset was used to train the
126         classifier to learn the difference between sick and healthy leaves.

127  For the classification tasks, the FFHQ dataset was split in train/validation/test sets of $70/15/15$, while the Plant-Village
128  retained a proportion of $70/20/10$.

---

[2]https://www.kaggle.com/jessicali9530/celeba-dataset

[3]This is a detail that was revealed through contact with the authors.

### 3.3 Hyperparameters

**Original research:** For the partial reproduction of Table 3 of the original paper, we limited ourselves to a sample of $n = 250$ images, rather than the $n = 1000$ randomly sampled images, as denoted in the Jupyter Notebook.

**Reimplementation:** The computational costs of training StylEx precluded an in-depth hyperparameter search. For all modules except the encoder, we found a learning rate of $2e - 4$ for the ADAM optimizer performs well, with $\beta_1 = 0.5$ and $\beta_2 = 0.9$. We found the training to diverge unless the encoder learning rate was lowered significantly to $1e - 5$. We ascribe this difference to the significantly smaller input size in our models, or subtle implementation differences in the original paper which we don't have access to.

The classifier used in the paper was MobileNetV1 [4], but we opted for a MobileNetV2 or ResNet-18. The authors asserted that the use of advanced networks identified more subtle cues from the datasets on the classification problems at hand, and for this purpose, we opted for ResNet-18. Additionally, we observed that the MobileNet model did not perform well on the CelebA dataset for gender classification on this image size. The components of the $\mathcal{L}_{rec}$ loss were scaled according to authors' suggestion in our correspondence: 0.1 for $\mathcal{L}_{rec}^x$ and $\mathcal{L}_{LPIPS}$, 1 for $\mathcal{L}_{rec}^w$. Other loss components were not scaled.

On the local GPUs, we used a batch size of 4 with 8 gradient accumulation steps, while we use a batch size of 16 with 4 gradient accumulation steps on the computing cluster. For the training of the MobileNet V2 classifier and the ResNet-18 classifier, we have set the learning rate to $lr = 1e - 4$, used a batch size of 128 and used the Adam [7] with default PyTorch parameters.

### 3.4 Experimental setup and code

We aimed to follow the experimental setup as closely as possible for our experiments. Our PyTorch implementation is available on GitHub[4] to further support and advance reproducibility in machine learning research. The repository provides explanations to run the described experiments.

### 3.5 Computational requirements

Our models were trained on three different machines, which were a 1) laptop NVIDIA RTX 3060, 2) an NVIDIA RTX 2070 Super and a 3) computing cluster containing GTX 1080 Ti GPUs. It must be noted that the first machine made use of the Windows operating system, while the latter two are Linux-based. For both the FFHQ dataset as well as the Plant-Village dataset, training was done until convergence, which was reached in 150K training steps for the FFHQ dataset and 260000 training steps on the Plant-Village dataset.

On the local GPUs, a batch size of 4 (RTX 3060) and 8 (RTX 2070 Super) was used alongside a gradient accumulation for 8 (RTX 3060) and 2 (RTX 2070 Super) steps. On the computing cluster, a batch size of 16 was kept, with a gradient accumulation parameter of 4. Depending on the hyperparameters of the batch size, and gradient accumulation, the computational time to run the experiments ranged between 20-50 GPU hours. Training for 150000 steps took 20 hours on an RTX 2070 Super.

## 4 Results

### 4.1 Results reproducing original paper

#### 4.1.1 Sufficiency

We calculate the percentage of flipped classifications after changing the top-10 attributes found by the *AttFind* procedure. The results can be seen in table 1. Our results on the author's model is within 1% of the accuracy reported in the paper. Our models perform show significantly worse performance on both perceived age (51% vs 93.9%) and plant healthiness (30% vs 91.2%), showing that the attributes discovered are not very relevant for classification.

---

[4]https://anonymous.4open.science/r/Explaining-In-Style-Reproducibility-Study-5665

|  | Ours |
| --- | --- |
| *Perceived Gender* | 94.8% |
| Perceived Gender (Model 1, $s = 2$) | 51% |
| Perceived Gender (Model 2, $s = 1$) | 21% |
| Plants ($s = 2$) | 30% |

Table 1: Percentage of flipped classifications on different datasets. Row in *italics* shows our experiment on the author's supplied model. $s$ represents the shift size used to generate the results. The shift sizes have been decided by qualitatively looking at the produced images.

### 4.1.2 Coherency and Distinctness

Similar to the original paper, we have conducted a user study ($n = 54$) to evaluate the distinctiveness of the found attributes and coherence of the generated images. The user study was divided into two parts - 1) a classification study and a 2) verbal description study, following a similar setup as presented in [16]. For the classification study, users are shown four animations in a grid format, each corresponding to a modification of a given attribute. In the verbal description study, the users were asked to look at four animations, and consequently describe in 1-4 words the changing attribute.

We have done this for the plant dataset as well as the FFHQ datasets. The order of the datasets was randomized to avoid biases and learning effects. All participants are undergraduate and graduate students who have some affinity and knowledge of machine learning. None of them have self-reported colourblindness. In A, a few examples can be found on the posed questions and the type of provided answers.

|  | **Wu *et al.*** | **Lang *et al.*** | **Ours** |
| --- | --- | --- | --- |
| Perceived Gender | 0.783 (±0.186) | 0.96 (±0.047) | Model 1: 0.52 (±0.2081) Model 2: 0.79 (±0.1599) |
| Plants | 0.91 (±0.081) | 0.916 (±0.081) | 0.66 (±0.323) |

Table 2: **User study results.** Partial reproduction of Table 2 of the original paper, on a subset of the datasets

Although our results seem to slightly outperform the results by Wu et al. (2021) on the perceived age classifier, it does not seem to outperform the method posed by Lang et al. (2021).

## 4.2 Results beyond original paper

To investigate the impact of attribute perturbation on the quality of the generated images, we compute the FID [3] between the original images and the generated images using [12]. We perturbed the images with increasingly more attributes in a cumulative fashion, starting from the 0th attribute which corresponds to only encoding and decoding the image. For the pretrained model from the original authors, we used the provided subset of 250 latents and their corresponding original images that were found in FFHQ. For our own models, we used subsets of 100 images (500 images for model 2) due to computational constraints with regards to running the *AttFind* algorithm. Our results, seen in 3, show that FID increases with the number of perturbed attributes. This result is not surprising, as changing an attribute can cause combinations of features not commonly seen in the original data distribution (e.g. young boy with lipstick). Moreover, for our reproducibility study, we noticed that perturbing more attributes at once, resulted in more artefacts, which also could have caused the FID to increase.

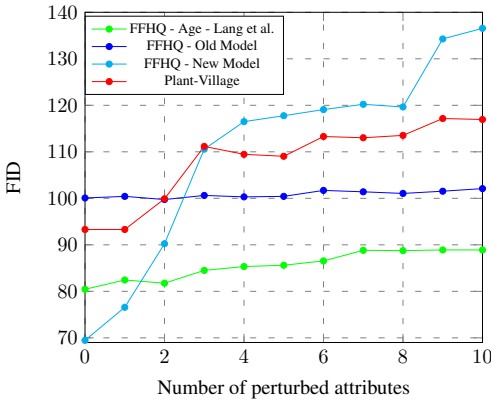

Figure 3: FID scores after perturbing top-$k$ attributes.

# 5   Discussion

Our experimental results support the claims posed in the original paper -
the attributes detected by StylEx are identifiable by humans to a certain degree, distinct and sufficient. However, due to the significantly lower resolution and poorer image quality of the models, these results are not comparable to the ones posed in the original paper.

**Reflection on our reproduction study** An important insight obtained during the conduction of the study is that the provided code did not cover the entire scope of the paper. Through a thorough study of both the code as well as the paper, we quickly noted discrepancies and missing elements that were fundamental - such as the network architecture, scaling of the losses and the hyperparameter configurations - to the original research.

We believe that researchers could enhance transparency and reproducibility in machine learning research by the addition of a reproducibility statement within their research, including the used hardware, software and details relevant to the proposed study (e.g. such as clarifications on the exact network architecture). Moreover, it is important to detail hyperparameter search spaces, final parameter settings for all the used architectures and baselines. We believe that transparency is fundamental to stimulate the large-scale deployment of machine learning algorithms.

## 5.1   What was easy

It was relatively easy to run the code as the provided Jupyter Notebook by the authors. The provided notebook was thoroughly documented and written in consistent coding styles, making the interpretation of the notebook easier. The provided notebook lacked the elements to fully reproduce the research; the training procedure of the network was missing, only one pretrained model was provided and four datasets were missing that we were required to add. As such, we had to partially re-implement the framework in PyTorch, while the original implementation was provided in TensorFlow. The addition of new datasets in our framework to accommodate the experiments was a relatively easy task.

## 5.2   What was difficult

Reproducing the experiments at the same computational scale as the authors was deemed to be the largest challenge, given the limited computational resources we had available. For the training of the model, the original authors made use of 8 NVIDIA V100s, which took the original authors a week to train at the full resolution of $256^2$, whereas we were restricted to the use of the computing cluster, Colab/Kaggle and our local GPUs. Due to this limitation, we had to scale down the resolution of the new images across the different datasets significantly. We scaled down the resolution of the generated images across the different datasets to a resolution of $64^2$, which limited the fidelity of the results. Additionally, we experienced the following issues with the original paper:

1. **Little to no hyperparameters were given in the paper**, e.g. on the scaling of the losses, the learning rates etc.

2. **Ambiguities about the training procedure**: the classifier in the notebook was trained on CelebA, instead of the FFHQ dataset, which we did not expect. This appeared to be a design choice by the authors, as the CelebA dataset contained labels, which the network could leverage information from. Additionally, softmax logits appeared to be added to the discriminator – which was not mentioned explicitly in the paper – but appeared to follow the cGAN [10] training procedure.

3. **Ambiguities on the network architecture**: It was not entirely clear what the dimensionality and the function was of the $Z$ vector, as the paper did not explicitly mention this.

4. **Ambiguities about the preprocessing pipeline of the images** before it enters the encoder/classifier - in contact with the authors, they appeared to scale the RGB values from $[-1, 1]$

The original authors did provide the hyperparameter configurations early on, which slightly reduced the time to explore the different possibilities, but the provided learning rate for example was too high for us. Additionally, the conversion of the *AttFind* algorithm from TensorFlow to PyTorch also proved to be a somewhat difficult exercise. The challenge predominantly laid on the integration of this algorithm within the new PyTorch codebase, which required a thorough understanding of the internal workings of the algorithm.

### 5.3 Communication with original authors

Three emails were sent to the first author of the paper. In these emails, we have asked for additional details of the proposed network architecture, hyperparameter configurations and the training procedure of the networks. These details were not noted in the paper, nor the provided code. Answers to these questions were provided promptly. Unfortunately, they were not able to share their code for the training procedure, as it contained too many internal dependencies from their perspective.

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

# A  User Study

## A.1  Classification Study

The participants were provided with the following instructions for the classification study:

- Look at the animations on the left. Both are examples of the same transformation (change in the image).
- Then look at the two candidates on the right, A (top-right) and B (bottom-right).
- Choose which one does a similar transformation to those on the left.

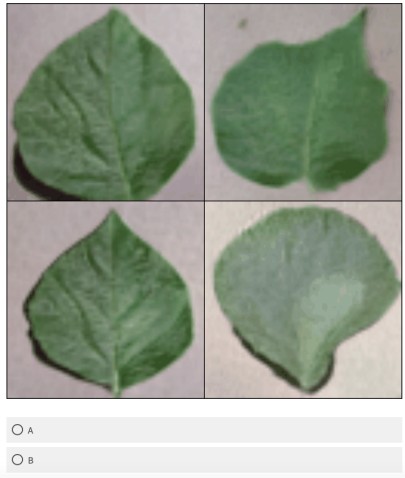

Figure 4: Sample question in the classification study, on the plants dataset.

**Correct answer:** B
**Accuracy:** 20/54 participants were correct.

## A.2  Verbal Description Study

The participants were provided with the following instructions for the verbal description study:

- Look at the animation.
- Describe in 1-4 words the single most prominent attribute that changes for all images.

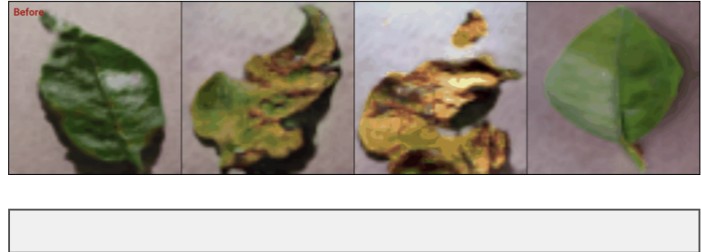

Figure 5: Sample question in the verbal description study, on the plants dataset.

**Users description:** lighting, colour/color, brightness, changes
**Most common word:** lighting

# B   Top attributes

## B.1   FFHQ - Model 1

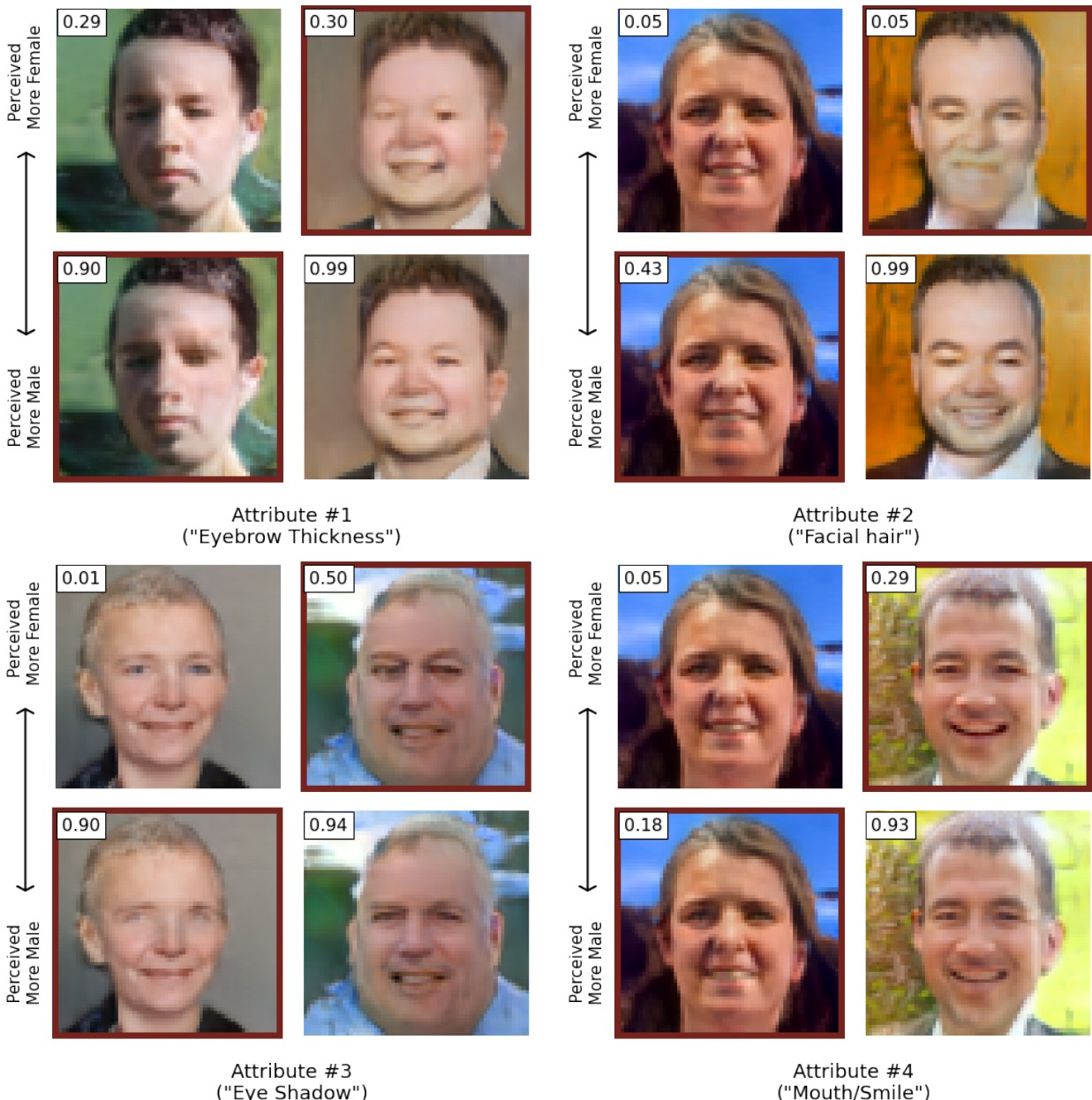

Figure 6: **Perceived Age - Model 1.** Classifier-specific interpretable attributes

 ## B.2   Plant Village

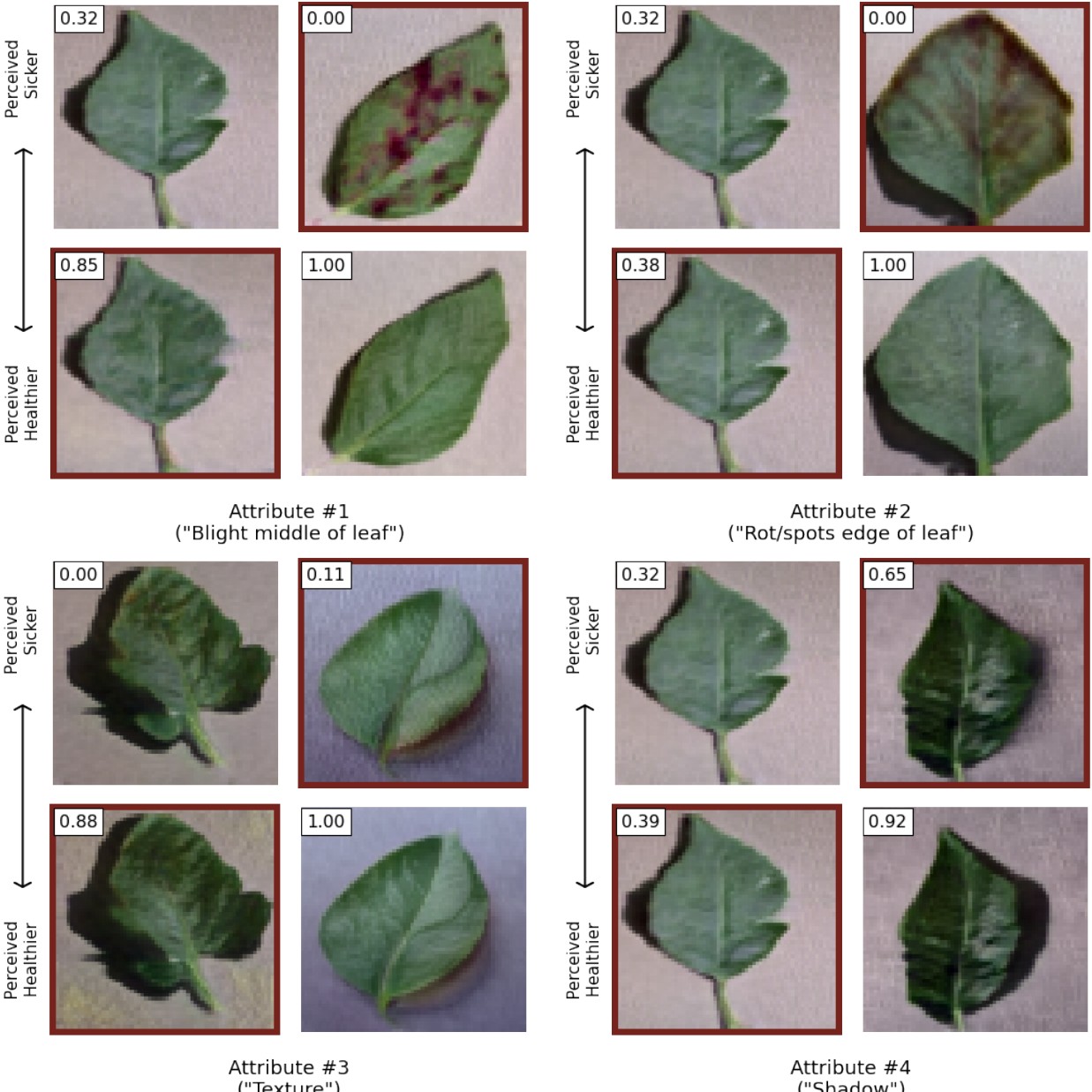

Figure 7: **Perceived Health.** Classifier-specific interpretable attributes

## B.3 FFHQ - Model 2

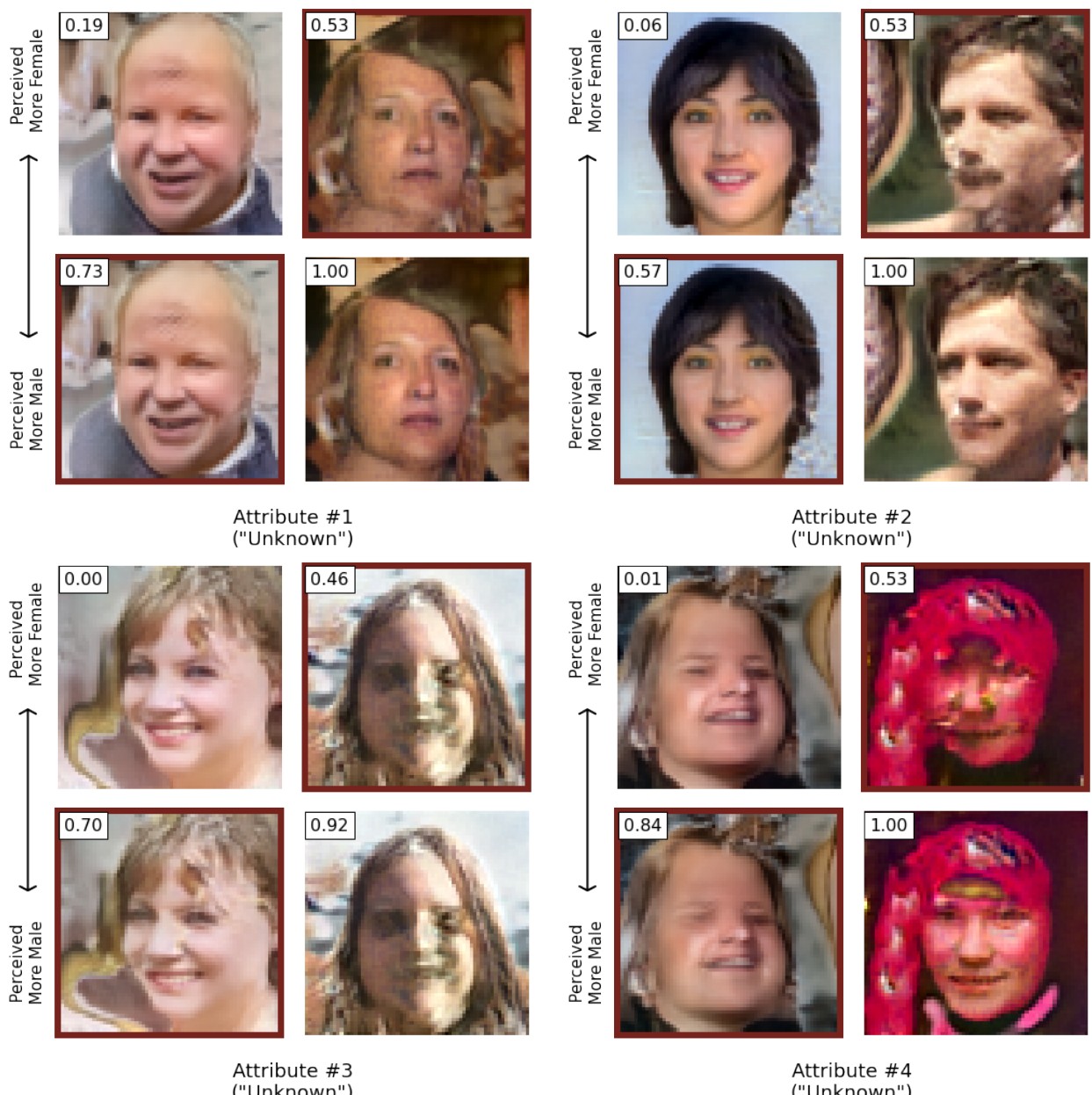

Figure 8: **Perceived Age - Model 2.** Classifier-specific interpretable attributes

