# OpenReview forum: "[Re] Explaining in Style: Training a GAN to explain a classifier in StyleSpace"
_ML_Reproducibility_Challenge/2021/Fall — RC2021_

### Official Review · Reviewer_i13Q · 2022-02-27
**Solid reproducibility effort**

**Rating:** 6
**Confidence:** 4

**Review:**

**Reproducibility summary:** The summary is overall clear and well summarizes the authors' reproducibility effort.

**Scope of reproducibility:** The authors check the original paper’s claims on three fronts (visual coherence, distinctness, sufficiency). For verifying visual coherence and distinctness, the authors repeat the original paper’s user study. For sufficiency, the authors repeat a selection of the original paper’s experiments, using a fewer number of datasets and lower resolution images due to computational constraints.

**Code:** The authors convert the original author’s Tensorflow code into PyTorch. The authors note that it was easy to run the original authors’ jupyter notebook, verify FFHQ results with the pretrained models, and extend the existing StyleGAN2 implementation.

**Communication with the original authors**: The authors have corresponded with the original authors several times via email to receive additional details about the network architecture, training procedure, and hyperparameters.

**Hyperparameter search:** The authors conduct a hyperparameter search and describe the process in detail. They note that hyperparameter search was difficult, partly due to underspecification in the original paper.

**Ablation study**: The authors did not conduct any ablation studies.

**Discussion on results:** The authors conclude that their experimental results are overall in support of the original paper’s claims. However, they got a significantly lower accuracy on the identification of the extracted attributes and sufficiency scores on their model. They note that their results are not directly comparable to the original paper’s due to the significantly lower resolution (64, original paper: 256) and poor image quality of the models.

**Recommendations for reproducibility:** The authors reflect on their experience and recommend future researchers to include a reproducibility statement detailing the used hardware, software, implementation details, and hyperparameters for transparaency and reproducibility in ML research.

**Results beyond the paper**: The authors compute the FID score, which is a metric commonly used to evaluate generative models, which compares the distribution of generated images with the distribution of real images used to train the generative model. They find that FID score increases with the number of perturbed attributes used to generate a counterfactual explanation. (A lower FID indicates better-quality images.) I found this to be an interesting additional analysis, but the authors don’t give an explanation for the results. I recommend adding a discussion. Does this align with their intuition or qualitative analysis results?

**Overall organization and clarity:** Overall, the report was organized and clearly written.

---

### Official Review · Reviewer_nGdB · 2022-03-17
**StylEX Review**

**Rating:** 8
**Confidence:** 4

**Review:**

This work came up with a new framework that explained the classification outcome of generative models. The work incorporates the classification result in the training phase of GAN itself to obtain better discernable feature. The work also attempts to provide explanations beyond the traditional grad CAM visualization, and gives insight how changing the important image attributes would lead to changes in classification output. The proposed method StyleEx does not rely on the image type, and the classifier.

We attempted to execute the codes according to the code provided by the authors. The code was understandable and usable. We found the results consistent with the results reported in the paper. However, the original work did not specify all the parameters, and other settings of the network. I believe since the authors caried out a lot of experiments on several datasets, the authors could not include all the network parameters and settings in the manuscript. The work conducted various experiments on 7 publicly available datasets. Because of time, and resource constraints we mostly verified the visual counterfactuals presented in the manuscript. The explanations and the counterfactuals suggested by the method made sense. However, the counterfactuals on gender classification produced some hard to believe results.

Overall, the work is very interesting, and significant. The code is well structured, easy to comprehend. The authors have clarified how to implement the codes, and mentioned about the dependencies. According to our analysis the reported results match with the experimental outcome, albeit deep learning approaches inherently contain some stochasticity. I believe if the authors can tabulate the hyperparameters, it will be easier for other researchers to implement the work.

---

### Meta-Review · Program_Chairs · 2022-04-09

**Recommendation:** Accept
**Confidence:** 5

**Metareview:**

Strong submission. A solid contribution to the reproducibility challenge.  The submission is accepted.

---

### Decision · Program_Chairs · 2022-04-09

**Decision:**

Accept

**Comment:**

Following the recommendation of reviewers and meta-reviewer, the paper is accepted for ML Reproducibility Challenge 2021, and will be published in the upcoming special edition of ReScience Journal.